# Brief Mindfulness Meditation Protects Chinese Young Women’s Body Image from Appearance-Focused Social Media Exposure: An Online Randomized Controlled Trial

**DOI:** 10.3390/healthcare14010120

**Published:** 2026-01-04

**Authors:** Xiaoxiao Zhang, Zixuan Zhang

**Affiliations:** Department of Communication Studies, School of Languages and Communication Studies, Beijing Jiaotong University, Beijing 100044, China; xxzhang1@bjtu.edu.cn

**Keywords:** body image, mindfulness meditation, appearance-focused social media, young women, body dissatisfaction, negative mood, Xiaohongshu (RED), brief intervention, randomized controlled trial, China

## Abstract

**Objectives**: Exposure to appearance-focused social media often leads to body image disturbance among young women. One promising intervention to lessen this negative impact is mindfulness meditation. This study aimed to determine whether a brief mindfulness meditation intervention could mitigate the adverse effects of exposure to appearance-focused social media content on body image and mood in young Chinese women. **Methods**: In an online randomized controlled trial, 168 women aged 18–35 years were randomly assigned to either an intervention group (*n* = 86) that listened to a ten-minute mindfulness meditation audio or to a control group (*n* = 82) that listened to a ten-minute recorded natural history text. After listening to the audio, participants viewed idealized body images on Xiaohongshu and compared themselves to these images. Outcome measures included state body dissatisfaction and negative mood. Data were collected at baseline (T0), post-intervention (T1), and post-exposure to images (T2). **Results**: At T0, groups did not differ in age, BMI, education, body dissatisfaction, or negative mood (all *p* > 0.05). From T0 to T1, both groups showed significant improvements in body dissatisfaction and mood. The intervention group’s scores decreased significantly (*p* = 0.008; *p* < 0.01), and the control group also showed significant improvements on both outcome measures (both *p* < 0.001). However, when exposed to the idealized images, only the intervention group maintained its improvements, with no significant change in body dissatisfaction or mood (*p* = 0.178 and *p* = 0.310, respectively) from T1 to T2, whereas the control group’s scores worsened significantly on both outcome measures (*p* < 0.001 for both). **Conclusions**: These findings suggest that even a brief mindfulness meditation intervention may buffer against the negative effects of idealized social media content on body image and mood.

## 1. Introduction

As of early 2025, over 5 billion people were active social media users, representing roughly 62.3% of the world’s population [1]. These social media platforms are dominated by highly curated visual content, which is now recognized as a key driver of appearance-related pressures [2]. Frequent exposure to appearance-focused social media tends to widen the perceived discrepancies between one’s own appearance and societal beauty ideals, thereby undermining body satisfaction and mood among young women [3].

The social media landscape in China differs significantly from that of Western countries. In China, western social media platforms such as Instagram and Pinterest are inaccessible due to government restrictions, which has created opportunities for domestic alternatives like Xiaohongshu to gain popularity and flourish [4]. Xiaohongshu blends features of Instagram and Pinterest, focusing on image-centric posts and aspirational lifestyle content. It now boasts around 300 million monthly active users, with young women forming the majority of its user base [5]. However, despite its popularity and aspirational content, Xiaohongshu has faced criticism for saturating users with idealized body images, raising concerns that these portrayals promote unrealistic beauty ideals and distort young women’s body-image perceptions.

Given that Xiaohongshu’s user demographic is predominantly female and that many of these users actively aspire to attain a ‘perfect’ face and figure, they are at heightened risk of these adverse psychological effects. In other words, young women who regularly compare themselves to the platform’s unattainable beauty ideals may experience deeper dissatisfaction with their own looks and even long-term mental health consequences. In this context, it becomes crucial to explore accessible, evidence-based interventions to lessen social media’s adverse effects on the body image of Chinese young women.

As a psychological intervention, mindfulness meditation effectively targets body-image difficulties while fostering emotional well-being [6,7]. Mindfulness involves intentionally regulating one’s attention on present-moment experiences with a nonjudgmental, accepting attitude [8,9]. A growing body of literature suggests that mindfulness meditation may serve as a protective factor against negative body image. For example, Delinsky and Wilson (2006) found that combining mindfulness training with mirror exposure significantly decreased participants’ emotional reactivity related to their body image after only three training sessions [10]. Similarly, another study found that women who practiced mindfulness during a swimsuit try-on session reported a less negative affect and reduced body dissatisfaction [11].

In recent years, mindfulness-based interventions that aim to improve body image have received increased attention in the literature. However, relatively few studies have examined whether mindfulness can function as a protective factor by buffering the detrimental impact of appearance-focused social media platforms. This underexplored topic indicates a notable gap in existing research, underscoring the novelty and importance of further investigation. Additionally, most existing studies prioritize multi-session mindfulness interventions to address women’s body image [12,13], whereas brief, one-off sessions remain relatively neglected. To date, only one study has specifically examined the efficacy of a brief, single-session mindfulness intervention. In that study, Atkinson and Diedrichs (2001) found that a 15-min video incorporating practical mindfulness techniques (e.g., thought decentering and mindful breathing) significantly enhanced positive internalization of appearance ideals, reduced sociocultural pressures related to appearance, and alleviated negative affect among female university students [14]. These preliminary findings highlight the promise of brief mindfulness interventions, given that their short duration makes them relatively easy to integrate into daily routines. They can offer rapid, in-the-moment relief during challenging situations. Accordingly, additional empirical work is warranted to evaluate the effectiveness of brief mindfulness interventions for body-image concerns.

As previously reviewed, evidence suggests that a brief, single-session mindfulness meditation can serve as an accessible, effective tool to lessen body-image distress triggered by upward comparisons to idealized Xiaohongshu images. However, no research to date has directly tested whether engaging in such mindfulness exercises can truly mitigate or prevent body-image distress prompted by these social media triggers. Therefore, the present study examined whether a ten-minute mindfulness meditation administered prior to Xiaohongshu use would reduce state body dissatisfaction and negative mood and buffer the adverse impact of subsequent exposure to idealized images. Given that Xiaohongshu is predominantly used by young women and that women are disproportionately affected by appearance-focused social media pressures [15,16], we focused exclusively on female participants. Specifically, we hypothesized that women who completed the mindfulness intervention would report lower immediate body dissatisfaction and a less negative mood than those in a control group, with effects expected to be small-to-moderate in magnitude. Furthermore, we hypothesized that the intervention group would maintain these improvements even after being exposed to idealized body images on Xiaohongshu, whereas the control group was expected to exhibit the typical increase in body dissatisfaction and negative mood in response to that exposure, with group differences again expected to be small-to-moderate in magnitude.

This paper is organized as follows: Section 2 and Section 3 describes the online randomized controlled experiment, including participant recruitment and eligibility screening, randomization and blinding, the mindfulness and natural-history audio conditions, the Xiaohongshu image selection and procedure with the forced appearance-comparison task, outcome measures, and statistical analyses. Section 4 presents descriptive statistics and the generalized linear mixed model findings for state body dissatisfaction and negative mood across baseline, post-intervention, and post-exposure assessments. Section 5 interprets the findings in relation to prior mindfulness and social media body-image research and the Chinese context, followed by Section 6, Section 7 and Section 8.

## 2. Materials and Methods

### 2.1. Study Design

We conducted an online randomized controlled experiment. Using purposive sampling, participants who met the eligibility criteria were recruited via advertisements posted on the first author’s personal Xiaohongshu account. Participants were randomly assigned to either the mindfulness intervention or a control group. Those in the intervention group listened to a ten-minute mindfulness audio, and those in the control group listened to a matched ten-minute natural-history recording. After completing their respective audio sessions, participants viewed idealized body images from a researcher-curated Xiaohongshu profile and subsequently compared themselves to these images. Outcome variables were measured at baseline (T0), post-intervention (T1), and post-exposure (T2). The study was conducted in accordance with the Declaration of Helsinki and was approved by Ethics Committee of School of Language and Communication, Beijing Jiaotong University (approval code: BJTU2025060501). This trial was registered in the Chinese Clinical Trial Registry (registration no. ChiCTR2500104893) on 25 June 2025. The randomized trial was reported in line with the CONSORT guidelines [17].

### 2.2. Participants and Recruitment

We performed an a priori power analysis using G*Power 3.1 to determine the required sample size. This analysis assumed a medium effect size (Cohen’s *d* = 0.5), a two-tailed test with α = 0.05, and a statistical power of 85%. This analysis indicated that a minimum of 79 participants per group was required. We then used purposive sampling to recruit individuals who met the inclusion criteria: (a) female, (b) 18–35 years of age, (c) stable internet access, (d) the ability to participate in the online experiment from a quiet, distraction-free environment (e.g., a private room); and (e) normal or corrected-to-normal vision without color-vision deficits (including color blindness or weakness). Participants were excluded if they: (a) had engaged in any cognitive or mindfulness training within the past year (including structured programs or self-guided practice); (b) had any diagnosed psychiatric disorder (past or present) or any chronic physical illness.

Participant recruitment began in early July 2025 via volunteer recruitment advertisements posted on the first author’s personal Xiaohongshu social media account. Each advertisement provided a brief overview of the study, including its objectives, the inclusion and exclusion criteria, and the requirements for participation. Interested individuals contacted the researcher via Xiaohongshu or WeChat to express interest and obtain additional study details. They were first directed to complete an initial eligibility screening through an online questionnaire, which collected demographic information and included questions addressing each inclusion and exclusion criterion. These eligibility questions were phrased neutrally and not explicitly labeled as “inclusion” or “exclusion” criteria in the questionnaire, thereby minimizing the likelihood that participants would infer the exclusion criteria and tailor their responses to appear eligible. Because the trial was conducted entirely online, all eligibility information was obtained via self-report. The researchers did not have access to participants’ medical records, so medical and psychological history criteria were verified solely through participants’ own disclosure. Individuals whose questionnaire responses indicated that they met all inclusion criteria were deemed eligible and were provided an online informed consent form to review and sign electronically. By providing consent, participants affirmed that they understood the study’s aims and their obligations, and acknowledged their right to withdraw at any point without any impact on their employment status.

A total of 202 individuals were initially enrolled. Twelve did not proceed to the trial: eight did not meet the inclusion criteria and four declined participation. This left 190 participants who were randomized to the intervention or control group. During the intervention phase, five participants in the intervention group did not complete the ten-minute mindfulness session. At the immediate post-intervention assessment, an additional two participants in the intervention group and five in the control group did not complete the outcome measures. At the final post-exposure assessment, a further two participants in the intervention group and eight in the control group were missing. Consequently, the post-exposure analyses included 86 participants in the intervention group and 82 in the control group. Participant flow through the study is illustrated in the CONSORT flow diagram (Figure 1).

The final sample comprised 168 women aged 17–35 years (intervention: M = 26.32, SD = 5.43; control: M = 26.63, SD = 5.17). BMI values ranged from 18.32 to 24.54 kg/m^2^, indicating a healthy weight range on average (M = 21.01, SD = 2.76 for intervention; M = 21.3, SD = 2.36 for control). In terms of education, only a small number of participants had less than a high school education (intervention group: 3 participants, 3.49%; control group: 2 participants, 2.44%) or only a high school education (intervention group: 12 participants, 13.95%; control group: 15 participants, 18.29%), while most participants held a bachelor’s degree (intervention group: 43 participants, 50%; control group: 39 participants, 47.56%), followed by smaller proportions holding master’s degrees (intervention group: 21 participants, 24.42%; control group: 20 participants, 24.39%) and doctoral degrees (intervention group: 7 participants, 8.14%; control group: 6 participants, 7.32%). Overall, these distributions indicate that the participants were relatively highly educated.

Participants were allocated in a 1:1 ratio to the intervention or control group. To preserve balance, block randomization with blocks of four was implemented. The allocation sequence was produced by an independent researcher using a computer-based random number generator. To maintain allocation concealment, the researcher responsible for generating the randomization sequence did not participate in recruitment or enrollment. To maintain blinding, the researchers remained unaware of the randomization sequence and group assignments, and participants were not informed of their assigned group throughout the trial.

### 2.3. Procedure

All participants provided informed consent. Prior to any study assessments, each participant joined a brief WeChat video call with a researcher, who verified that the testing environment was quiet and distraction-free (e.g., a private room with no other people present). Eligible participants received the online questionnaire via WeChat, which was designed and administered using the Credamo online survey platform (www.credamo.com). Participants first completed baseline VAS measures of state body dissatisfaction and negative mood (T0; approximately 3 min). Immediately thereafter, participants completed the assigned 10-min audio task via Credamo: those in the control group listened to a natural history recording, whereas those in the intervention group listened to a guided mindfulness meditation. Immediately after the audio task, participants completed the post-intervention VAS measures (T1, approximately 3 min). Participants then completed the image-viewing and forced appearance-comparison task (approximately 10 min). Specifically, participants viewed a set of idealized images (see Image Selection for details) and were instructed to compare specific body parts of the woman depicted with the corresponding parts of their own bodies. Participants indicated whether each of nine body areas (thighs, arms, buttocks, waist, hips, biceps, breasts, legs, and stomach) was “much smaller,” “smaller,” “about the same size,” “larger,” or “much larger” than the corresponding area on the woman. In addition, they evaluated the attractiveness of their own face and their overall physical appearance relative to the woman in the image, indicating whether they were “much more attractive,” “slightly more attractive,” “about as attractive,” “less attractive,” or “much less attractive.” Such forced social comparison tasks were widely employed in prior research investigating body image responses to idealized media images [18,19,20,21]. In the present study, this approach enabled us to examine whether the mindfulness intervention could mitigate any body distress stemming from young women’s exposure to appearance-focused social media content. Immediately after this task, participants completed the post-exposure VAS measures (T2, approximately 3 min) and then provided demographic information. Thus, the time between completion of the T0 and T1 was 10 min (the audio task), and the time between completion of T1 and T2 was approximately 10 min (the image-viewing/comparison task), with no scheduled delay between phases. Each VAS assessment required approximately 3 min.

### 2.4. Interventions

We replicated the mindfulness meditation and control task scripts from Fraser et al. [22]. The script for the audio recording was provided by the original study’s authors. It was then adapted for use in the current study. Specifically, two independent bilingual experts were engaged to carry out the translations. The script was back-translated into English by a separate set of bilingual researchers to ensure accurate capture of the original meanings. Subsequently, two experts in mindfulness meditation reviewed the translated text. Their goal was to ensure that Chinese participants could fully understand all content and details in both audio recordings. Any discrepancies identified were addressed by the researchers. This procedure ensured semantic equivalence and cultural suitability of the final versions for the Chinese context [23]. Both audio recordings were made by the same individual (an instructor certified in the Mindfulness Intervention for Emotional Distress [MIED] program, Peking University Continuing Education, 2024 [24]) and were of equal task length (i.e., ten minutes).

#### 2.4.1. Intervention Group

Participants assigned to the intervention group completed a ten-minute guided meditation designed to enhance present-moment awareness. During the session, they were encouraged to focus on their breath and other bodily sensations, such as sounds, sights, and touch. The meditation encouraged an attitude of acceptance and nonjudgment toward these sensations. Participants were specifically guided to observe their thoughts and physical sensations without evaluation throughout the session. Notably, this exercise was not centered on body image. Previous research has demonstrated that listening to this mindfulness recording significantly enhances state mindfulness levels in participants, as compared to a control group [25].

#### 2.4.2. Control Group

Participants assigned to the control group listened to a ten-minute audio recording of the same natural history passage used by Fraser et al. [22]. The recording provides factual descriptions of natural landscapes and animals and does not include any appearance- or body-related content. In prior studies, this passage has been described as emotionally neutral and mildly relaxing for young women [22,26].

### 2.5. Image Selection

After completing one of the interventions outlined above, participants viewed 12 idealized body images sourced from public Xiaohongshu accounts. Image selection followed a structured two-stage procedure informed by prior research [25,27]. In the first phase, 30 images were identified using the hashtags #PerfectBody, #BodyChallenge, #AppearanceIcon, #FitnessGoddess, and #FullBodyShot. These hashtags were purposefully selected to capture portrayals of women consistent with dominant beauty ideals in China. In the second phase, after consulting with three expert researchers in body-image research, the initial set of 30 images was refined to 20. The selection was made according to key criteria: each image needed to have clearly visible facial features, a recognizable body type, and a full-body presentation. Subsequently, twenty young women aged 18–35 were recruited through the Xiaohongshu account of the first researcher to pretest the 20 images. They rated the images based on five dimensions using 5-point Likert scales (see Table 1): (1) Emotion (5 = Very good, 1 = Very bad), (2) Perception (5 = Very good, 1 = Very bad), (3) Arousal (5 = Very excited, 1 = Not excited at all), (4) Body Anxiety (5 = Always, 1 = Not at all), and (5) Social Comparison (5 = Always, 1 = Not at all). This method was previously employed by Li et al. [28] to assess the idealization of female images from social media, demonstrating strong evaluation efficacy, which is why we adopted it for the current study. The mean score for each dimension was calculated for every image. A hierarchical ranking was then applied to derive the final set of 12 images, giving primary weight to Body Anxiety and Social Comparison, followed by Perception, Emotion, and Arousal, to maximize stimulus engagement and relevance. These images were then presented to participants through a curated, simulated Xiaohongshu profile specifically created for this study. All visible social feedback (e.g., ‘like’ counts and comments) was concealed to ensure that each image was viewed in a realistic context without extraneous social cues. All 20 pre-tested images are presented in Appendix A; the 12 images selected for the experimental stimulus set are indicated.

### 2.6. Outcome Measures

The questionnaire used in this study consisted of two main sections. The first section collected demographic characteristics (age, educational level) along with self-reported height and weight to calculate body mass index (BMI; kg/m^2^). The second section assessed participants’ state body dissatisfaction and mood using Visual Analogue Scales (VAS) [29]. These VAS assessments were administered at T0, T1 and T2. Participants indicated their current feelings by marking a point along a horizontal line anchored from 0 (not at all) to 100 (very much). Body dissatisfaction was evaluated using the averaged responses to three items: “fat,” “physically attractive” (reverse-coded), and “satisfied with your body size” (reverse-coded). These VAS items primarily assess dissatisfaction related to body weight and shape as well as body size (e.g., perceived “fatness” and dissatisfaction with body size), alongside global evaluations of physical attractiveness; this content aligns with thinness-oriented concerns commonly reported among young women. Higher scores indicated greater dissatisfaction with one’s body at that moment. Mood was assessed by averaging responses to five items: “depressed,” “happy” (reverse-coded), “anxious,” “angry,” and “confident” (reverse-coded). Higher scores reflected a more negative mood state. Prior studies indicate that VAS provides reliable and sensitive measure of changes in body satisfaction and mood in young women, making it well suited to pre–post experimental designs [8,30,31]. The VAS has also demonstrated good convergent validity and internal consistency, specifically among young adult women in China [5]. In the present study, both mood and body dissatisfaction scales exhibited high internal consistency, with Cronbach’s alpha coefficients of 0.89, 0.87, and 0.90 for mood, and 0.95, 0.96, and 0.93 for body dissatisfaction at T0,T1 and T2, respectively.

## 3. Statistical Analyses

Statistical analyses were conducted using IBM SPSS Statistics for Windows, Version 25.0 (IBM Corp., Armonk, NY, USA).Descriptive statistics were used to summarize participant characteristics and outcome variables and are reported as mean (SD) for continuous variables and n (%) for categorical variables. Baseline group differences were examined using independent-samples t tests (continuous variables) and Pearson’s χ^2^ tests (categorical variables). Normality of the outcome variables at each time point (T0, T1, and T2) was assessed using the Shapiro–Wilk test.

Primary analyses were conducted using generalized linear mixed models (GLMM) to examine whether changes in state body dissatisfaction and negative mood over time differed by condition. Two separate models were specified to examine the effects on each of these two outcome variables. To measure differences in the rate of change from T0 to T2, time was coded into two dummy variables. The first dummy variable captured the contrast between T0 and T1 (T0 coded as 1, T1 as 0); the second dummy variable captured the contrast between T1 and T2 (T1 as 0, T2 as 1). Condition (Intervention group vs. Control group) and its interaction with time were included as predictors in each model. This analytical approach enabled the examination of the rate of change in the outcome variables over time, as well as differences in this rate between groups. To further investigate the interaction between condition and time, simple effects of time within each condition were examined. The condition variable was re-coded in two ways (first coding the control group as 0 and the intervention group as 1, then reversing the coding), so each group served as the reference category in one of the analyses. This design enabled a separate evaluation of within-group changes in both the intervention and control groups. Effect sizes for these changes were calculated as Cohen’s d. Estimates of the fixed effects from these models were reported below.

## 4. Results

### 4.1. Preliminary Results

Normality assumptions were examined for the outcome variables at each time point. Shapiro–Wilk tests indicated no evidence of substantial departures from normality in either group (all *p* > 0.05). Baseline characteristics were comparable between the intervention and control groups, with no statistically significant differences in age (t = −0.38, *p* = 0.63), BMI (t = −0.75, *p* = 0.27), or education level (χ^2^ = 0.38, *p* = 0.98). Similarly, baseline body dissatisfaction (t = −0.15, *p* = 0.88) and negative mood (t = −1.72, *p* = 0.09) did not differ significantly between groups. Descriptive statistics (mean ± SD) are presented in Table 2.

### 4.2. State Body Dissatisfaction

For state body dissatisfaction, the GLMM analysis indicated that the Condition × Time interaction was not significant in the initial phase (T0 to T1; b = 0.28, *p* = 0.935, Cohen’s *d* = 0.02). In the later phase (T1 to T2), although the effect was in the expected direction, it did not reach statistical significance (b = −6.48, *p* = 0.062, Cohen’s *d* = −0.41). Within the intervention condition, state body dissatisfaction decreased significantly from T0 to T1 (b = −6.45, *p* = 0.008, Cohen’s *d* = −0.51), with no significant change from T1 to T2 (b = 3.27, *p* = 0.178, Cohen’s *d* = 0.21). The control condition showed a similar initial improvement, with a significant decrease from T0 to T1 (b = −6.17, *p* = 0.013, Cohen’s *d* = −0.59). However, unlike the intervention condition, it exhibited a significant rebound in body dissatisfaction from T1 to T2 (b = 9.74, *p* < 0.001, Cohen’s *d* = 0.61). See Figure 2.

### 4.3. Negative Mood

For negative mood, the GLMM revealed no significant Condition × Time interaction during the initial period (T0 to T1: b = 1.61, *p* = 0.378, Cohen’s *d* = 0.18). However, a significant Condition × Time interaction was observed in the subsequent period (T1 to T2: b = −6.69, *p* < 0.001, Cohen’s *d* = −0.77). In the intervention condition, negative mood decreased significantly from T0 to T1 (b = −6.16, *p* < 0.001, Cohen’s *d* = −0.71). This reduction in negative mood was maintained, with no significant change observed from T1 to T2 (b = 1.3, *p* = 0.310, Cohen’s *d* = 0.15). In the control condition, negative mood significantly decreased from T0 to T1 (b = −4.55, *p* = 0.001, Cohen’s *d* = −0.52), followed by a significant rebound from T1 to T2 (b = 7.99, *p* < 0.001, Cohen’s *d* = 0.92). See Figure 3.

## 5. Discussion

Appearance-focused social media platforms, such as Xiaohongshu, perpetuate unrealistic beauty standards within Chinese society. Because idealized social media images exert substantial influence on beauty norms among young Chinese women, there is an urgent need for effective psychological interventions to ameliorate the associated physical and mental health risks. To address this critical need, the present study investigated whether a brief audio-based mindfulness meditation, administered immediately before exposure to idealized body images on Xiaohongshu, could attenuate subsequent increases in body dissatisfaction and negative mood. Significantly, this is the first study to test a mindfulness-based intervention specifically addressing social media-related body-image concerns in young Chinese women.

We hypothesized that a mindfulness meditation session would lead to significantly lower levels of state body dissatisfaction and negative mood after the intervention, compared to the control task. However, the results only partially supported this hypothesis. As anticipated and consistent with previous research [8,32,33], participants in the mindfulness group showed significant reductions in state body dissatisfaction and negative mood following the intervention. However, participants in the control group, who listened to a ten-minute audio recording of a natural history-text, also showed significant short-term reductions in these outcomes. Notably, both the mindfulness and control conditions led to comparable immediate improvements in body dissatisfaction and negative mood after their respective audio tasks. This finding contrasts with some prior research that reported minimal or no improvement in neutral control tasks [34,35], yet aligns with the findings of Gobin, McComb, and Mills (2022) [27]. In their study, Gobin et al. found no significant between-group differences in weight and appearance dissatisfaction when comparing a self-compassion writing task to a neutral word-sorting exercise. They attributed these null findings to nonspecific factors, such as habituation, which may also explain the results of the current study. Specifically, repeated exposure to the experimental procedures may reduce participants’ self-consciousness over time, thereby potentially lowering reported body dissatisfaction as participants become more comfortable with or less sensitive to the experimental context.

We also hypothesized that mindfulness meditation would act as a protective factor against body image distress triggered by exposure to idealized body images on Xiaohongshu. This hypothesis was supported by the results. Participants in the intervention group maintained lower levels of state body dissatisfaction and negative mood even after completing a forced appearance-comparison task with idealized body images on Xiaohongshu. In contrast, this protective effect was entirely absent in the active control group. Participants in this group listened to a ten-minute audio recording of a natural history text and experienced a significant increase in both outcomes. To clarify why this brief intervention remained effective under forced appearance comparison, we draw on evidence from prior research. Specifically, we consider several interrelated psychological mechanisms, including decentering, attentional control, emotion regulation, and self-compassion [36]. Specifically, mindfulness interventions cultivates present-moment attentional focus [37] and a decentered perspective [38], allowing individuals to disengage from automatic upward appearance comparisons commonly triggered on image-centric social media platforms like Xiaohongshu and to view negative body-related thoughts as transient mental events rather than self-defining truths [39]. This cognitive distancing (decentering) reduces ruminative self-criticism [40,41], while the mindfulness intervention’s nonjudgmental stance simultaneously nurtures emotion regulation and self-compassion which enables acceptance of distressing feelings without avoidance and directly counters the harsh self-evaluation and shame that underlie body dissatisfaction [42]. Together, these mechanisms blunt the emotional and cognitive impact of forced appearance comparisons, with mindful individuals exhibiting less post-exposure decline in mood and body satisfaction in the face of idealized social media images compared to the control condition.

In Western contexts, a growing body of experimental work, largely conducted with Instagram-based thinspiration or fitspiration stimuli, has similarly examined whether brief, audio-guided mindfulness practices can offset the body-image harms of idealized social media exposure. For example, Hooper et al. (2024) [6] found that a single 10-min mindfulness meditation ameliorated the negative effects of viewing idealized social media images on self-esteem, mood, and body appreciation in young women, relative to an active control. At the same time, not all micro-interventions appear equally robust when participants encounter idealized body imagery online. Fraser et al. (2022) [22] reported that although brief auditory interventions (including mindfulness meditation) produced immediate reductions in state body dissatisfaction, these benefits did not persist following subsequent exposure to idealized body images. Taken together, prior Western findings suggest that brief guided mindfulness practices can be beneficial, yet their protective effects may be vulnerable under more challenging exposure conditions. Our findings demonstrated that even a brief, single-session mindfulness exercise could serve as an immediate and accessible tool for mitigating social media-related harm to body image among young Chinese women [14,28,43]. However, as we noted earlier, the efficacy of short-term mindfulness interventions remains inconsistent. Some studies suggest that its benefits depend on moderators such as one’s motivation to practice, the length of the session, and the frequency of practice [44]. For example, greater practice frequency has been linked to higher positive affect and sustained well-being [42]. Indeed, regular mindfulness practice cultivates a present-moment focus that improves attentional regulation and emotional awareness. This, in turn, reduces maladaptive automatic responses [43,44]. Given that our intervention was a brief, one-off session, it is unlikely to generate long-term buffering effects against appearance-focused social media content.

Furthermore, cultural factors may influence the generalizability of the present findings. In China, collectivistic values and Confucian-heritage norms place a strong emphasis on relational harmony and social evaluation, which can make physical appearance and body-image concerns especially consequential in people’s everyday interactions. In addition, the concerns about social evaluation and “face” (mianzi) can further heighten sensitivity to how one is perceived by others, which may intensify the emotional impact of upward appearance comparisons on image-centric social media platforms. Cross-cultural evidence supports this possibility. For example, compared with Croatian women, Chinese women report higher thin-ideal internalization and greater perceived pressure from family, friends, and the media to conform to prevailing beauty norms, and they show a greater increase in body dissatisfaction following thin-ideal priming [45]. Relatedly, research with Chinese college students suggests that social media use is associated with stronger upward social comparison, which in turn predicts appearance anxiety—defined as stress in response to others’ evaluations of one’s appearance [46]. Taken together, these culturally patterned appearance pressures may moderate both the magnitude and durability of brief mindfulness related buffering effects. Therefore, while our findings suggest that core mechanisms proposed in Western mindfulness literature (e.g., present-moment attention and decentering) can operate in a Chinese, Xiaohongshu-based context, future cross-cultural replications should directly test these cultural moderators and establish measurement equivalence across settings.

## 6. Implications

### 6.1. Theoretical Implications

This study addresses a significant gap by providing randomized controlled trial evidence that a brief, audio-guided mindfulness meditation can buffer against social media-induced body dissatisfaction and negative mood among young Chinese women using Xiaohongshu. Research to date has largely been conducted with Western samples, leaving the dynamics of these psychological processes in non-Western populations underexplored. Our findings provide crucial Chinese data, demonstrating that the protective mechanisms of mindfulness are robust and transferable to a distinct sociocultural context. This is particularly significant given the high prevalence of body image concerns among young women in China, which are often driven by intense media pressure to conform to the unattainable beauty ideals found on appearance-focused social media platforms. Meanwhile, as discussed above, culturally patterned appearance pressures and social-evaluation concerns in China may condition the magnitude and durability of these brief buffering effects [43,44]. Accordingly, our findings provide crucial evidence from China while highlighting the importance of testing cultural moderators when translating brief mindfulness micro-interventions across platforms and societies.

### 6.2. Practical Implications

The most significant practical implication of this study is the validation of a highly accessible, low-cost, and scalable mental health tool that can fundamentally change how young women engage with social media. As a brief, audio-guided online exercise, the intervention can be easily self-administered immediately before or even during social media use, serving as a form of “digital first aid” for emotional well-being. This means that rather than lapsing into mindless scrolling or making automatic negative comparisons, individuals can shift into a more mindful mode of social media use by pausing or reflecting when they encounter distressing triggers. For example, a young woman who begins to feel anxious while viewing an idealized body image, a brief mindful pause may increase the likelihood of disengaging from the feed, softening self-evaluative thoughts, or reframing the comparison rather than continuing to scroll and amplifying distress. By supporting this momentary shift in attention and appraisal, brief mindfulness practice may help attenuate spikes in state body dissatisfaction and negative mood following exposure to idealized appearance content. Thus, the intervention equips users with an immediate coping mechanism, transforming their social media engagement from passive, emotionally vulnerable consumption into active, self-aware use.

The scalability of this mindfulness intervention further amplifies its potential impact at the community or even population level. Because the exercise is brief, low-cost, and easily disseminated online, it can be integrated into daily routines and scaled up to reach a wide audience with minimal barriers. Indeed, at the public health level, such an intervention represents a promising preventive strategy that could be implemented on a broad scale. Educational institutions and community organizations could introduce these brief mindfulness techniques in digital literacy programs or wellness curricula, equipping students with proactive coping skills for navigating the online world from the outset. By normalizing mindful engagement practices in school and university settings, institutions can help safeguard young people against social media’s adverse effects before those effects take root, potentially reducing the need for reactive interventions or counseling down the line.

Finally, these findings have direct implications for social media platforms and the technology industry in promoting healthier user engagement. There is growing recognition that social media companies share responsibility for users’ mental well-being and should take active steps to mitigate potential harms. Our study suggests that one practical step is to integrate mindfulness support tools directly into the user experience. Platforms like Xiaohongshu or Instagram could, for instance, incorporate optional “mindful break” prompts into their interfaces. After a period of continuous scrolling, or when the platform detects potentially triggering content such as a series of idealized body-image posts, it could cue users to pause and engage in a short guided breathing exercise or reflection. This feature would be analogous to the “take a break” reminders already available in some apps, but specifically tailored to help users recenter emotionally. By incorporating the kind of brief intervention validated in this study, social media platforms would provide users with an immediate tool to manage their reactions, thereby taking a proactive role in reducing the distress that platform content might inadvertently cause. Such measures could not only assist users in the moment but also signal a broader shift in platform design toward prioritizing user well-being. In embracing these strategies, technology companies would demonstrate an alignment of their operations with emerging mental health best practices and acknowledge the importance of moderating social media’s impact on user behavior.

## 7. Limitations and Future Research

This study has several limitations that warrant consideration. First, although the visual stimuli were designed to resemble typical Xiaohongshu posts and were presented remotely, the fixed 10 min viewing time for each image may not reflect participants’ natural viewing behavior. This limitation, in turn, could diminish the ecological validity of the results. Future studies should strive for more naturalistic designs that better mirror authentic user experiences. For example, participants could be allowed to freely navigate a simulated or real social media feed rather than being limited to fixed viewing times. Such an approach would help determine whether the observed effects on body dissatisfaction and mood hold true under more ecologically valid conditions.

Second, our outcome measures relied on self-reported ratings of state body dissatisfaction and mood, assessed using visual analogue scales (VAS). Such self-report measures are inherently susceptible to response biases. Participants’ answers may have been influenced by demand characteristics or by transient mood states at the time of reporting [47,48]. For instance, if participants became aware of the study’s purpose, they might respond in a socially desirable way. Additionally, an experimentally induced mood can “spill over” into self-evaluations that are unrelated to the actual content of the questions [49]. Such biases could also distort the true effect sizes [50]. To address this limitation, it would be beneficial for future studies to include additional multi-modal assessment methods alongside self-reports. Ecological Momentary Assessment (EMA) is one promising approach. By capturing participants’ effect and body image experiences in real time during their daily lives, EMA can track naturally occurring fluctuations and context-dependent changes that survey questionnaires might miss [51,52]. Incorporating objective behavioral indicators would also provide observable evidence to corroborate self-reports. For example, researchers could track participants’ social media usage patterns, record appearance-checking behaviors, or monitor engagement in appearance-related activities. Moreover, adding physiological measures could objectively index participants’ emotional reactions with less risk of subjective bias. For instance, heart rate variability and skin conductance have been used to measure stress or emotional arousal in body image experiments. By measuring such physiological responses in future mindfulness intervention studies, researchers can determine whether mindfulness attenuates the body’s stress reactions to idealized images, such as reductions in autonomic arousal, even when self-reported mood and body dissatisfaction are subject to bias. In summary, employing a combination of self-report, EMA, behavioral observations, and physiological measures would allow for cross-validation of outcomes. This multi-method approach would mitigate the limitations of any single measure, providing a more robust and valid assessment of how brief mindfulness interventions influence state affect and body image. Each modality offers a different window into the participant’s experience, and convergent evidence across these modalities would greatly strengthen confidence in the findings.

Third, the intervention in our study was limited to a single brief mindfulness session with immediate post-test measurement and no follow-up. This design prevents any conclusions about the longevity of its benefits. Without additional sessions or later assessments, we could not determine whether the mindfulness-induced improvements in mood and body image would persist or attenuate over time. To address this issue, future studies should incorporate multiple follow-up assessments over time to evaluate the durability of the effects and whether the initial benefits are maintained or even amplified. For instance, follow-up measurements could be conducted several weeks or months after the session. In addition, repeated or extended mindfulness practice might be necessary to counteract the long-term influence of ongoing social media exposure. In particular, we recommend implementing longitudinal, smartphone-delivered interventions where mindfulness sessions are integrated into participants’ regular social media use over several weeks. For example, a short mindfulness exercise could be scheduled before or after each day’s social media browsing. Such a design would not only test the immediate buffering effects of mindfulness in a more natural context, but also reveal any cumulative or lasting changes in body dissatisfaction and mood resulting from sustained mindfulness practice. A mobile-based approach could increase ecological validity by integrating the intervention into participants’ everyday routines, and it would allow researchers to monitor how mindfulness exercises perform in real-world settings over time.

Forth, the sample consisted exclusively of female participants. Although women typically report higher levels of body dissatisfaction, men also experience significant body image concerns [53]. Future studies should evaluate brief mindfulness interventions in male samples, taking into account gender-specific manifestations of body dissatisfaction. For example, men’s body dissatisfaction more often centers on muscularity, leanness, and height [54,55]. Beyond gender, it is important to test the intervention in more diverse populations to determine whether its protective benefits generalize across different groups. Adolescents are one such critical group, as the negative impact of social media on body image and mood is especially pronounced during adolescence and young adulthood [56,57]. For example, studies could focus on adolescent girls who actively engage with social media to see how they respond to the mindfulness intervention. In addition, individuals with elevated body image concerns or eating disorder symptoms represent another key population that might benefit from mindfulness techniques to build psychological resilience.

By addressing these limitations, future research can build on our results to more conclusively determine the scope and durability of mindfulness as a protective tool against the adverse effects of social media on body image. The above recommendations, which include increasing ecological validity, extending the intervention duration, broadening sample diversity, and adopting multi-method assessments, will advance our understanding of when and for whom brief mindfulness interventions are most effective. Ultimately, implementing these strategies will guide the development of evidence-based programs to promote a healthier body image in the digital age.

## 8. Conclusions

This study demonstrated that, compared to a control condition, a brief ten-minute mindfulness meditation effectively mitigated the subsequent increases in state body dissatisfaction and negative mood induced by exposure to appearance-focused social media. Given the pervasive nature of idealized imagery on Xiaohongshu and similar social platforms, encouraging young women to engage in a brief mindfulness exercise before exposure could offer immediate protective benefits. It may also help improve resilience against appearance-related distress. Future research should explore how ongoing mindfulness practice can counteract the long-term negative impacts of repeated exposure to such content.

## Figures and Tables

**Figure 1 healthcare-14-00120-f001:**
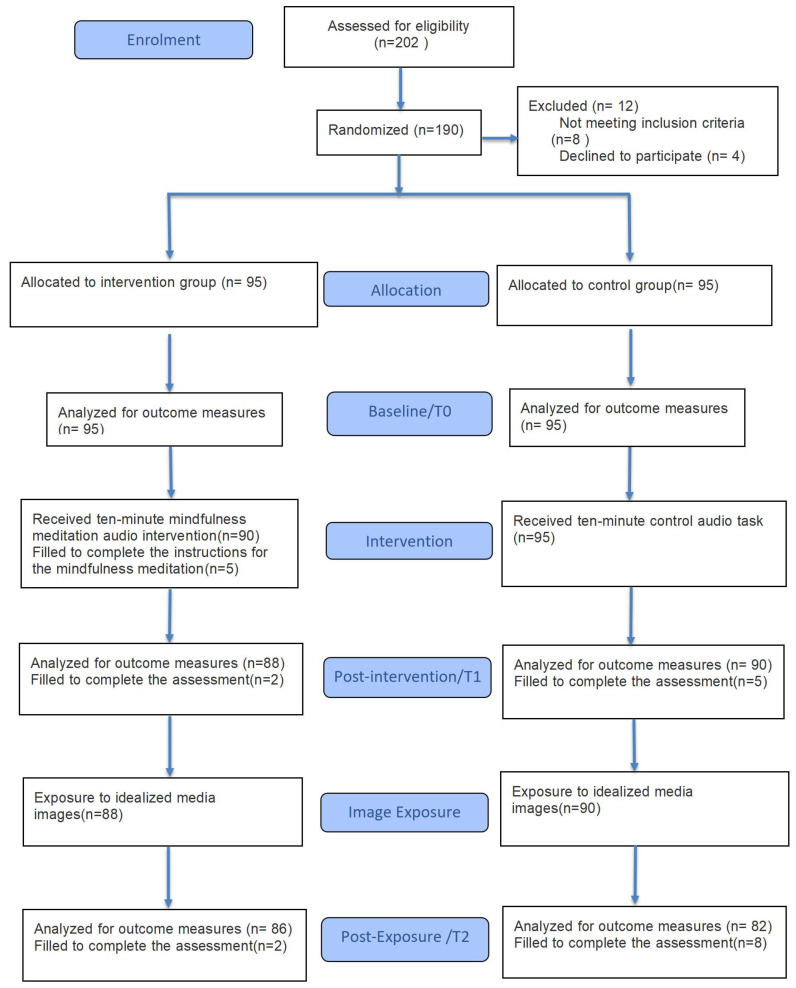
Participant flow through the randomized clinical trial.

**Figure 2 healthcare-14-00120-f002:**
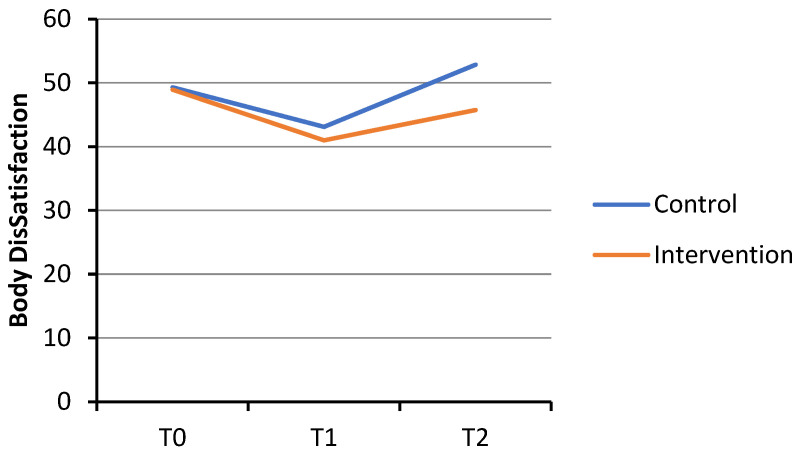
Changes in state body dissatisfaction scores over time. Note. State body dissatisfaction scale ranged from 0 to 100.

**Figure 3 healthcare-14-00120-f003:**
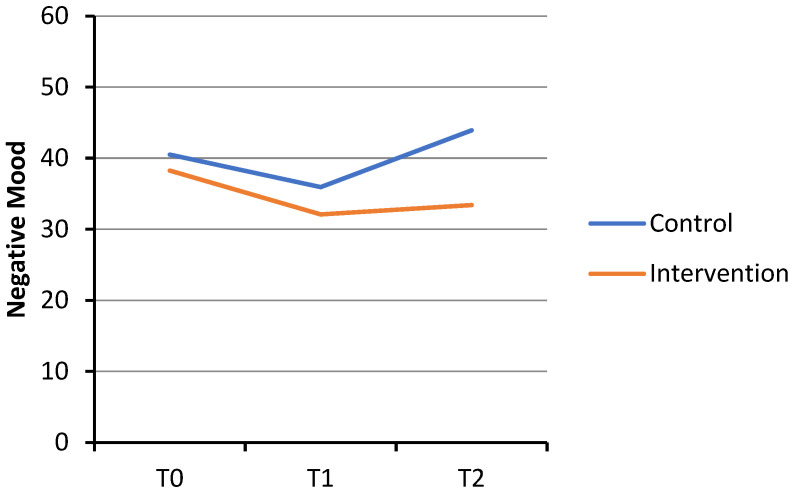
Changes in negative mood scores over time. Note. Negative mood scale ranged from 0 to 100.

**Table 1 healthcare-14-00120-t001:** Rating scores of 20 Xiaohongshu images by 20 participants volunteers.

Images	Perception	Emotion	Arousal	Social Comparison	Body Anxiety
1	4.45	4.30	3.70	2.40	1.95
2	4.30 ^a^	3.60	3.45	2.25	1.75
3	4.45 ^a^	3.50	3.25	2.10	1.75
4	4.35 ^a^	3.50	3.50	2.30	1.55
5	4.85	4.65	4.05	1.70	1.40
6	4.90	4.80	4.55	2.05	1.55
7	4.55	4.45	4.05	1.55	1.40
8	4.30 ^a^	4.25	3.85	1.30	1.10
9	4.75	4.60	4.30	1.85	1.40
10	4.95	4.90	4.70	2.2	1.90
11	4.65	4.55	4.25	1.75	1.35
12	4.20 ^a^	4.00	3.70	1.15	1.05
13	4.10 ^a^	3.95	3.65	1.25	1.15
14	4.70	4.60	4.30	1.80	1.40
15	4.30 ^a^	4.20	3.90	1.45	1.10
16	4.50	4.40	4.05	1.7	1.35
17	4.65	4.60	4.20	1.85	1.40
18	4.20 ^a^	4.10	3.80	1.35	1.05
19	4.80	4.75	4.35	1.90	1.50
20	4.35	4.15	3.75	2.40	1.70

Note. ^a^ Not included among the stimuli used in the formal experiment.

**Table 2 healthcare-14-00120-t002:** Descriptive statistics (mean ± SD) for body dissatisfaction and negative mood at baseline (T0), post-intervention (T1), and post-exposure (T2), by group.

Outcome	Time	Intervention (*n* = 86) Mean ± SD	Control (*n* = 82) Mean ± SD	Total (*n* = 168) Mean ± SD
Body Dissatisfaction	T0	48.92 ± 15.88	49.28 ± 15.93	49.10 ± 15.85
	T1	42.47 ± 17.40	43.11 ± 13.08	42.78 ± 15.40
	T2	45.74 ± 16.56	52.86 ± 16.00	49.21 ± 16.63
Negative Mood	T0	38.26 ± 9.12	40.50 ± 7.70	39.35 ± 8.51
	T1	32.10 ± 9.46	35.95 ± 9.14	33.98 ± 9.47
	T2	33.39 ± 8.60	43.93 ± 8.16	38.54 ± 9.89

## Data Availability

The original contributions presented in the study are included in the article; further inquiries can be directed to the corresponding author. The data presented in this study are available on request from the first author due to privacy restrictions.

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
