# Peer review of "Brief Mindfulness Meditation Protects Chinese Young Women’s Body Image from Appearance-Focused Social Media Exposure: An Online Randomized Controlled Trial"

_healthcare, 2026, doi:10.3390/healthcare14010120_

Round 1
Reviewer 1 Report
Comments and Suggestions for Authors
I reviewed the manuscript "Brief mindfulness meditation protects Chinese young women’s body image from appearance-focused social media exposure: An online randomised controlled trial". I must appreciate the authors' hard work in following the methodology with its true spirit.
- The flowchart explains the process of selection and assignment very clearly
- The formal procedure of consent taking was adopted through WeChat
- Random assignment was followed by blinding
However, some limitations are found in the research. For example,
- VAS is a scale based on self-understanding. Recording bias is an issue.
- Normality of the data was not tested
- Though the data collected was not fully continuous, most statistical techniques used to analyze it give better results with continuous data. however, it is norm as plenty of similar literature is available.
The sample was selected in a random manner, but the authors wrote that a purposive sampling scheme was used. Also, the sample size needs to be determined for the probability sampling.
Reviewer 2 Report
Comments and Suggestions for Authors
This study attempts at drawing a direct connection between mindfulness training - e.g., via recorded meditation - and body-image respectively the positive or negative perception of one's own body among young female Chinese citizens when confronted with highly curated physical appearances on social media. The study design and and procedure is well detailed and correspondingly can be easily replicated, possibly on a larger scale and with gender differentiation.
For additional more concrete suggestion for improvement, please see attached file.

Reviewer 3 Report
Comments and Suggestions for Authors
Thank you for your timely and interesting research. I hope my suggestion will help to improve your paper.
The introduction provides a clear overview of the context and the research gap. However, it would be useful to condense the introduction by summarizing key points more succinctly. I suggest providing more specific hypotheses, such as the expected magnitude of change in body dissatisfaction and mood.
Also please enhance the discussion of the practical implications of the findings, such as the potential impact on social media usage behavior. Besides, I strongly suggest providing a more detailed exploration of the psychological mechanisms behind mindfulness's protective effects. As you mentioned in Section 6.1, it would be great to include a broader comparison to similar studies in both Western and non-Western contexts. Thus, please improve your discussion with a non-Western context, for example, how cultural factors might influence the generalizability of the findings.
I recommend proposing a detailed plan for future research, including longitudinal studies and interventions targeting diverse populations.
Images used in the study could be presented as supplementary materials.
Reviewer 4 Report
Comments and Suggestions for Authors
This is a clearly written study. Please find the comments:
- Please add more keywords to improve indexation. Up to ten.
- p-value should be italicized.
- The introduction seems good. It is unclear why this study focused only on females. Please add several justifications. Overall, indicate in the title of the paper that this was a female sample. Please specify "gender‐specific manifestations of body dissatisfaction".
- Please specify how the inclusion and exclusion criteria were checked (140-147 and 152-154) as this study was conducted online. How was this "initial eligibility screen" done? Based on what information? Self-reported?
- Please specify "a passage from a natural history text, which was described as neutral yet relaxing".
- I would recommend presenting 20 images used in this study in Supplementary Materials.
- Descriptive statistics for baseline, post-intervention and post-exposure values of negative mood and body satisfaction should be presented.
- Please indicate the time between baseline, post-intervention and post-exposure.
- Please indicate mean ± SD near body dissatisfaction and negative mood.
- Figure 2: Sometimes the authors use the term body dissatisfaction and sometimes body satisfaction. Please specify. It seems that Y axis in this figure was indicated incorrectly as satisfaction.
